# Waterpipe Use among Adolescents in Germany: Prevalence, Associated Consumer Characteristics, and Trends (German Health Interview and Examination Survey for Children and Adolescents, KiGGS)

**DOI:** 10.3390/ijerph17217740

**Published:** 2020-10-22

**Authors:** Stephanie Klosterhalfen, Daniel Kotz, Benjamin Kuntz, Johannes Zeiher, Anne Starker

**Affiliations:** 1Institute of General Practice, Centre for Health and Society, Addiction Research and Clinical Epidemiology Unit, Medical Faculty of the Heinrich-Heine-University, 40225 Duesseldorf, Germany; StephanieKatharina.Klosterhalfen@med.uni-duesseldorf.de (S.K.); Daniel.Kotz@med.uni-duesseldorf.de (D.K.); 2Robert Koch Institute, Department of Epidemiology and Health Monitoring, General-Pape-Str. 62-66, 12101 Berlin, Germany; KuntzB@rki.de (B.K.); ZeiherJ@rki.de (J.Z.)

**Keywords:** KiGGS, waterpipe, shisha, smoking, Germany, adolescents

## Abstract

Waterpipe (WP) use is popular among youth worldwide, but epidemiological data from Germany are scarce. We aimed to describe prevalence rates of WP use (current, last 12 months, ever) and analysed correlates and trends among 11- to 17-year-olds in Germany. Analyses were based on data from the “German Health Interview and Examination Survey for Children and Adolescents” study during 2014–2017 (*n* = 6599). Changes in WP use prevalence compared with 2009–2012 were used to describe trends. Associations with sociodemographic characteristics and cigarette smoking were assessed with multivariable logistic regression models. Prevalence of current WP use among adolescents was 8.5% (95% confidence interval (CI) = 7.5–9.6), use in the last 12 months was 19.7% (95% CI = 18.3–21.2), and ever use was 25.8% (95% CI = 24.2–27.5). High prevalence rates were particularly found among 16–17-year-olds. During 2009–2012, these prevalence rates were 9.0%, 18.5%, and 26.1%, respectively. WP use was associated with older age, male sex, migration background, lower educational level, and current smoking status. Among current WP users, 66.2% (95% CI = 60.0–71.9) identified themselves as non-smokers, and 38.1% (95% CI = 32.5–44.0) had used WP ≥ three times in the last month. WP consumption is popular among German youth, and prevalence rates have not changed over time. Specific prevention strategies to reduce harmful WP consumption among youth should be implemented.

## 1. Introduction

In recent decades, there has been a worldwide increase in the prevalence of waterpipe (WP) use among young people. Historically, the popularity of WPs spread from India, across continents, until its consumption became accepted in the Western world as an alternative form of tobacco smoking. Regular consumption of WPs by broad sections of the population is a phenomenon that was not observed prior to the end of the 20th century [1,2,3].

Although the name (hookah, shisha, narghile, argileh, boory, goza, or hubble bubble), size, and design of WPs vary from region to region, they all function in the same way. The characteristic of this time-consuming (average duration of 47 min) method of tobacco smoking is that the smoke passes through water before being inhaled into the lungs once cooled [4]. Figure 1 shows the required components of a WP. To fill the tobacco head, a special, mostly sweet and flavored WP tobacco called maassal can be used, as well as alternative tobacco-free products such as steam stones [5]. To heat the tobacco, WP charcoal (or alternatively, an electronic heat source [6]) is used. See: Figure 1. Components of a waterpipe.

WP use differs from conventional cigarette use not only with respect to the length of a smoking session; WP tobacco tastes often sweeter due to the added flavors, and the inhalation of cooled smoke seems less irritating to the mucosae and lungs. In addition, WPs can be smoked as a group, e.g., at a party, and can therefore create a social experience [7].

These aspects provide insight into why WP use is popular among adolescents, why many WP users do not perceive themselves as “conventional smokers” [8], and why some users underestimate the health risks of WP consumption [9,10,11,12]. First experiences with the consumption of tobacco typically take place during the period of experimentation during adolescence. The most frequent first tobacco product tried by young people is the cigarette (followed by cigar, smokeless tobacco, and WP) [13]. During this period, adolescents are at special risk of developing dependency, and the risk of early deterioration of health increases [14]. Different cultural and socioeconomic backgrounds as well as use of other tobacco products can be determinants regarding the consumption of WP by adolescents [15,16,17]. The aromatic taste of WP tobacco (e.g., apple, cherry, melon) appeals to young people and can be associated with a more pleasant, longer smoking experience which leads to increased nicotine exposure and dependence potential [18,19,20]. Furthermore, the consumption of WP is associated with other harmful health effects similar to those associated with cigarette smoking [21]. In addition to the increased risk of carbon monoxide poisoning, which can result from combustion of the WP charcoal [22], smoking WP can cause acute to chronic impairment [23], negative impacts on executive brain function, or carcinogenic changes in various organs including the lungs and cardiovascular system [24,25,26]. Sharing a WP among different people can also increase the risk of transmission and infection with bacterial or viral diseases [27], which is particularly relevant during times of acute pandemic such as the present global novel coronavirus disease (COVID-19) pandemic.

The number of shisha bars (almost 6000) and the consumption of WP tobacco have risen in Germany [28]. The increasing number of WP cafés can influence societal acceptance, and these serve as a place of social exchange for adolescents, just like pubs in former generations [12,29].

In Germany, there are legislative measures at the both state and the federal level to regulate WP consumption (Bundesnichtraucherschutzgesetz (“Federal Non-Smoker Protection Act”), Jugendschutzgesetz (“Youth Protection Act”), Tabakerzeugnisgesetz (“Tobacco Products Act”), Nichtraucherschutzgesetz (“Non-Smoker Protection Act”)). The German Tobacco Products Act regulates ingredients, emission levels and information requirements for tobacco and related products. In 2016, the ingredients of WP tobacco changed (% content of glycerin). The Youth Protection Act regulates the distribution of tobacco products. In 2007, the age limit for the consumption of tobacco products in public has been raised from 16 to 18 years. It is not permitted to sell tobacco products to minors. Children and adolescents under the age of 18 are not allowed to smoke in publicly accessible rooms in places open to the public and otherwise in public places. These measures were accompanied by a tobacco prevention program. Purchase of WP tobacco and accessories or the entry to a shisha bar are not permitted to people under 18 years of age.

Apart from regional studies, there are only a few population-based studies on the prevalence of WP consumption among adolescents in Germany. The German Health Interview and Examination Survey for Children and Adolescents (KiGGS) study and studies of the Federal Centre for Health Education (BZgA) such as the Drug Affinity Study have collected data on awareness about and use of WP, differentiated according to migration background, frequency of consumption, and combined consumption of tobacco cigarettes, WPs, e-products, and tobacco heaters [16,30]. National and international study findings indicate that male adolescents or youth with a migration background use WP more often than girls or people without a migration background [3,16,30]. Regarding socioeconomic or educational factors, there seems to be a relationship between WP use and lower educational levels in Germany, whereas international studies have reported opposite findings [3,16,30]. A study by the German health insurance DAK (“DAK-Präventionsradar”) has collected prevalence figures of WP consumption among school children [31]. Prevalence rates of 6–14% for current and 22–44% for ever use of WPs are reported for adolescents in Germany under 18 years of age [16,27,30,31,32]. Regarding international prevalence rates, current WP consumption varies widely, from 2.2% in Romania to 36.9% in Lebanon [15,33]. Several studies from the United States (US) reported increasing rates of WP use among 11- to 18-year-olds between 2009 and 2017 [34]. Smoking a WP is a common form of tobacco use among adolescents in the US [12].

However, little is currently known about the factors associated with WP use. The influence of a one- or both-sided migration background, the socioeconomic status (SES) of the family, and sex, have not yet specifically been investigated in Germany. Data are also missing on the percentage of WP users who perceive themselves as smokers or non-smokers. This is an important issue, which can influence the perception of health risks of WP tobacco consumption and the creation of prevention programs. We, therefore, aimed to evaluate WP use and associated factors among German adolescents. More specifically, based on data of the second wave of the German Health Interview and Examination Survey for Children and Adolescents (KiGGS Wave 2), in the present study, we aimed to (i) investigate the prevalence of WP consumption among 11- to 17-year-old boys and girls; (ii) describe the frequency of WP use and the self-assessed smoking status; (iii) examine the associations between sociodemographic factors, smoking status and WP consumption among adolescents; and (iv) to monitor trends between the previous and the current wave of the KiGGS study.

Due to a large study sample, the KiGGS study—in contrast to other population-wide studies conducted in Germany—allows the surveillance of prevalence figures more detailed (e.g., one- or both-sided migration backgrounds, survey of 11-year-olds, survey 12-month prevalence) and to include statements on self-assessed smoking status. These data can help in the identification of different risk profiles to develop targeted group-specific and gender-sensitive prevention strategies.

## 2. Materials and Methods

### 2.1. Methods

The KiGGS study is part of health monitoring conducted by the Robert Koch Institute (RKI) on behalf of the Federal Ministry of Health in Germany. KiGGS focusses on health status, health behavior, living conditions, protective and risk factors, and healthcare among children, adolescents, and young adults living in Germany. Cross-sectional data have been collected at three time points: the KiGGS baseline study (2003–2006), KiGGS Wave 1 (2009–2012) and KiGGS Wave 2 (2014–2017). The response rate (according to AAPOR response rate 2) of KiGGS Wave 2 was 40.1% in total [35]. A multi-step approach was used to include people with a migration background in KiGGS Wave 2. The share of children and adolescents of non-German nationality in KiGGS Wave 2 corresponds to the population figures from the Federal Statistical Office [36]. The concept, methodology, and analyses of KiGGS are described in detail elsewhere [35,37,38,39].

Comparable to the KiGGS baseline study, respondents for KiGGS Wave 2 were selected randomly based on the population registers of 167 representative German municipalities and cities (two steps sampling process). The study population of KiGGS Wave 1 consists of re-invited participants from the baseline study supplemented by newly invited children aged 0–6 years. KiGGS Wave 2 (like KiGGS baseline study) was comprised of an interview and examination part, whereas KiGGS Wave 1 was conducted as a telephone interview survey [37,38,39]. To achieve an optimal number of respondents and sample composition, a variety of measures were applied (e.g., phone calls or home visits) [35,36], resulting in a total of 15,023 respondents aged 0–17 years. The analyses of WP consumption were restricted to data from 11- to 17-year-old respondents (*n* = 6599), collected using a written questionnaire. To identify trends in comparison with the previous wave, the results from Wave 1 were compared with the currently collected prevalence rates from Wave 2. The study was approved by the ethics committee of Hannover Medical School (No. 2275-2014).

### 2.2. Measurements

The prevalence of WP use was assessed with the question “Have you ever smoked a waterpipe or shisha?” Respondents who affirmed having used a WP were defined as “ever WP user” and were further asked “Have you smoked a waterpipe or shisha in the last 12 months?” (yes defined as “last-12-month WP user”) and “If you think about the last 30 days, on how many days did you smoke a waterpipe or shisha?” (response options: ≥1 day, defined as “current user” or “None in the past 30 days”). Regarding the frequency of use during the past month, we classified responses according to one, two, or ≥three times.

To determine the SES of the family, an index was generated based on information of the parents’ level of education, occupational status, and income (equivalized disposable income). Thus, respondents were classified as belonging to a family with “low”, “medium”, or “high” SES [40]. School type was surveyed by asking the parents “Which type of school does your child go to?”, with nine response options: “Primary school”, “Secondary school”, “Middle school”, “School with secondary and middle educational program”, “Integrated comprehensive school”, “Academic secondary school”, “Technical secondary school”, “Special school”, and “Other”. Due to its federal structure, there is no uniform school system in Germany. As some federal states now have a two-tier school system, we categorize for the following analyses, two groups for secondary school: “Secondary/Middle/Comprehensive school” and “Technical/Academic secondary school”. Young people who no longer attended school were assigned to the corresponding category based on the highest level of education they achieved [41].

To assess migration background, all respondents were asked about their own and their parents’ country of origin: “In which country were you born?” and “In which country were your parents born?” A one-sided migration background meant that one parent was not born in Germany or had no German citizenship; a both-sided migration background meant that the child himself/herself migrated to Germany and had at least one parent who was not born in Germany or both parents were born abroad [42].

To assess smoking status, respondents were asked “Do you currently smoke?”, with the following response options: “No”, “Daily”, “Several times a week”, “Once a week”, or “Less than once a week”. All respondents who answered in the affirmative were defined as a “current smoker”. The data collected for KiGGS Wave 2 are available from the RKI Research Data Center (https://www.rki.de/EN/Content/Health_Monitoring/Public_Use_Files/public_use_file_node.html).

### 2.3. Statistical Analyses

The descriptive analyses of WP use patterns (current, last 12 months, ever) stratified by sociodemographic characteristics and smoking status are presented, differentiated for female and male respondents, as percentages with 95% confidence intervals (CIs). Weighting with regard to age, sex, federal state, German citizenship, and the child’s parents’ level of education was applied to ensure representative data for children and adolescents living in Germany. Comparison of current prevalence figures and those obtained between 2009 and 2012 was based on descriptive statistics and is presented as percentage with 95% CI. Three multivariable logistic regression models were applied to explore associations between different WP use patterns and sociodemographic characteristics and smoking status for girls and boys: model I = current WP use vs. never WP use, model II = last 12 month WP use vs. never WP use and model III = ever WP use vs. never WP use.

Respondents with missing data were excluded from the regression analyses. Data were analyzed using Stata 15.1 (Stata Corp., College Station, TX, USA). Stata’s survey procedures were applied to account for the clustered sampling design.

## 3. Results

The prevalence of current WP use among 11- to 17-year-old adolescents in Germany was 8.5% (95% CI = 7.5–9.6; *n* = 446) in the period 2014–2017. Almost every fifth adolescent had used WP within the last 12 months (19.7%, 95% CI = 18.3–21.2; *n* = 1101), and 25.8% (95% CI = 24.2–27.5; *n* = 1415) were ever WP users (weighted data). Table 1 presents prevalence of different WP use patterns, sociodemographic characteristics, and smoking status, stratified by sex (weighted data, missing data regarding SES (*n* = 145), education (*n* = 710), migration background (*n* = 33), and current smoking status (*n* = 852)). The pattern of missing values showed a higher amount of missing values among boys with migration background, boys with lower SES and lower education level, and among girls with lower SES and multivariable analyses showed that the odds of missing values are especially high among boys with a both-sided migration background (data not shown). Boys were more likely than girls to report current (10.6% vs. 6.3%), last 12-month (22.1% vs. 17.3%), and ever (28.1% vs. 23.4%) WP use. Respondents with a migration background and current smokers reported using WP more often than those without a migration background or non-smokers (current, last 12 month, and ever WP use). For example 17.9% (95% CI = 11.0–27.8) of the boys with a one-sided migration background compared to 9.7% (95% CI = 8.2–11.4) of the boys without a migration background and 46.8% (95% CI = 37.2–56.7) of male current smoker compared to 8.2% (95% CI = 6.7–10.0) of currently non-smokers, reported current WP use. In the age group 17 years, 50.6% of girls (95% CI = 44.2–56.9) and 61.6% of boys (95% CI = 54.9–67.9) reported having ever used a WP in their lifetime. Of the 17-year-old girls 12.9% (95% CI = 9.6–17.1) reported current WP use and 33.8% (95% CI = 28.2–40.0) in the last 12 months. For boys these rates were 26.4% (95% CI = 20.9–32.8) and 50.3% (95% CI = 43.5–57.2), respectively.

Among adolescents who used WP in the 30 days before the survey (respondents: *n* = 446), 37.9% (95% CI = 32.0–44.1) reported use on a single day, 24.1% (95% CI = 19.1–29.9) reported having smoked WP on two days and 38.1% on three or more days (95% CI = 32.5–44.0). Regarding self-assessed smoking status among current WP users, 30.1% (95% CI = 23.3–37.8) of girls and 40.3% (95% CI = 31.4–49.9) of boys considered themselves smokers.

The results of the three multivariable regression analyses regarding sociodemographic characteristics, current smoking status, and WP consumption are presented in Table 2. The adjusted odds ratios (ORs) of current vs. never WP use (model I) were higher in adolescents with older age and current smokers (girls: OR = 1.97, 95% CI = 1.69–2.29 and OR = 48.27, 95% CI = 24.12–96.59; boys: OR = 2.20, CI = 1.92–2.52 and OR = 67.57, 95% CI = 18.02–253.32). Concerning migration background, we found that boys with a one-sided migration background used WP more often than boys without a migration background (OR = 3.03, 95% CI = 1.36–6.77). We found similar associations when comparing WP use in the last 12 months vs. never WP use (model II). In addition, girls with a both-sided migration background showed a lower OR for WP use than girls without a migration background (OR = 0.38, 95% CI = 0.22–0.65), and girls with a lower educational level showed a higher OR for WP use than girls with higher educational levels (OR = 1.82, 95% CI = 1.32–2.51). We also found the above-mentioned associations when comparing ever vs. never WP use (model III), except that the adjusted OR was also higher in girls from a family with low SES compared with girls belonging to a family with high SES (OR = 1.66, 95% CI = 1.02–2.71).

The prevalence rates of WP use in Wave 2 (2014–2017) were similar to those identified earlier in Wave 1 (2009–2012), as shown in Figure 2.

## 4. Discussion

Among German 11- to 17-year olds surveyed in the period 2014–2017, 8.5% reported being current WP users and about 26% reported being ever users. The use of WP seems to be most common in the age group of 16–17-year-olds. A considerable proportion (62%) of current WP users had smoked a WP twice or more in the last month. Only one-third of WP users considered themselves smokers. We found positive associations of WP use with older age, male sex, and current smoking status. Regarding the associations between WP consumption and education level or migration background, an inverse relationship was observed for both genders in some analyses. As shown in Table 2, the association between lower educational level and the use of WP was more pronounced among girls, whereas the association between the migration background and the use of WP is found primarily among boys. The prevalence rates did not differ much from those obtained during 2009–2012.

The prevalence rates found in the present study are highly congruent with data collected in 2015 by the BZgA [43]. The two nationwide surveys yielded comparable prevalence rates for both current use (KiGGS Wave 2 (11- to 17-year-olds): 8.5% vs. Drug Affinity Study (12- to 17-year-olds): 8.9%) and ever use (25.8% vs. 27.3%). The prevalence identified in the European School Survey Project on Alcohol and other Drugs (ESPAD) in Austria was strikingly higher: in 2019, 21% of 14- to 17-year-olds reported current and 51% reported ever WP use [44]. A possible reason for the difference may be the difference in age groups, as prevalence increases with age. This may also explain the comparatively high prevalence rates for young people (14- to 16-year-olds) living in the German state of Bavaria (current WP use: 20.1%; ever use: 48.9%) who also participated in the ESPAD study in 2015 [45]. Comparing the prevalence rates reported in Germany with those in the US (2009–2017), nationally representative estimates indicate lower prevalence figures (current use among high school students (grades 9 to 12): 4.8%, ever use: 14.3%); however, representative state-wide estimates showed comparable figures (current use: 11.6%; ever use: 22.5%) [34].

Within the present study we aimed to explore the frequency of current WP consumption and self-assessed smoking status. Most current WP users reported a WP use frequency of no more than twice in the past 30 days. This consumption pattern is also seen in previous studies [32,46]. Reasons for the difference in consumer behavior, for example, in comparison with (daily) cigarette consumption, could be owing to the inflexibility of the stationary tobacco use method and its time-consuming nature. Most current WP users identified themselves as non-smokers. Thus, WP consumption is not perceived as smoking, a result which has been also reported elsewhere [8].

The results of the KiGGS study showed variation in WP use according to sex, age, migration background, and current smoking status. We found higher ORs for current and ever use among respondents who were male, older, and who had a one-sided migration background (boys). These findings are in line with prior national and international studies [30,33,34,47]. Migration background is a known correlate of WP use described in previous KiGGS Waves and other studies. Whereas in the first Wave of the German Health Survey for Children and Adolescents (KiGGS Wave 1, 2009––2012) [48], boys with a both-sided migration background were found to use WPs more often (current and ever) than those without a migration background, we found counterintuitively low prevalence for WP use with a both-sided migration background but high prevalence with a one-sided migration background among boys in KiGGS Wave 2. Hence, we speculate that the particular high amount of missing values among boys with a both-sided migration background might explain their low prevalence of WP use. The case of girls was reversed; we found an association of a both-sided migration background and lower ORs for current and ever WP use. Similar results have been described for smoking adults (over 18 years) in Germany [49]. Our findings also point out that young people who regard themselves as current smokers were up to 68 times more likely to use WP than non-smokers. Associations regarding this kind of dual use have been reported in other studies [33,47].

Concerning the trend in prevalence figures, our study found stable figures over time. For 12- to 17-year-old boys, the BZgA reports similar trends in the figures for current WP use. First, these figures decreased from 2007 (16.3%) to 2011 (9.8%), but then remained at this level until 2015. For 12- to 17-year-old girls, a similar trend can be seen over time. The prevalence figures for the current use of WP ranged from 7.4% (2011) to 6.4% (2015) [50].

The present study entails the following limitations. Owing to the cross-sectional design, it was not possible to make conclusive statements about causality with respect to the results. Responses given in KiGGS Wave 2 are self-reported data, which are always associated with biases. Respondents may remember of the corresponding answer categories inaccurately (recall bias) or may give socially acceptable answers (social desirability bias). As there are different terminologies for WP, the use of pictures within the questionnaire would probably have been preferable to ensure that all respondents have the same understanding of the tobacco product. To be able to assess the health risks arising from the consumption of WP tobacco, the ingredients of WP tobacco play an integral part. Unfortunately, the composition of WP tobacco or the number of puffs during a session could not be investigated in this study. Over the course of the KiGGS study, there has been a change in methods: KiGGS Wave 2 was conducted using self-report questionnaires and KiGGS Wave 1 using telephone interviews, which are more susceptible to socially desirable response behavior [51]. As with all surveys, the possibility of bias owing to selective non-participation also exists. It is assumed that people who participate in a health study also have greater health awareness and therefore differ from the general population in terms of smoking behavior (selection bias) Systematic identification of patterns of missing items was not feasible, but could help to interpret results more accurately in further studies.

The described selection effects were partially corrected by weighting. Thus, the observed results may be generalizable to 11- to 17-year-olds in Germany, which is a strength of the KiGGS study. Furthermore, it was possible to identify different WP consumption patterns (current, in the last 12 months, and ever), the frequency of WP consumption, and the combined consumption of tobacco cigarettes and WPs, as well as the association of WP use with sociodemographic characteristics and cigarette smoking status.

## 5. Conclusions

Older age, male sex, migration background, lower educational level, as well as current cigarette smoking were found to be associated with WP use among German adolescents. WP consumption is popular among adolescents but does not seem to have increased substantially in recent years. Continuous monitoring of trends in prevalence and use behavior is important to yield an evidence basis for developing targeted group-specific and gender-sensitive prevention approaches within public health prevention strategies. In addition to preventive programs within schools, it would be also useful to provide information about the health hazards and addiction of WP use in sports clubs or on preferred social networks (e.g., YouTube, Facebook, Twitter) visited by young people. A targeted gender-specific approach could also be made here. The law for the protection of youth, which has been adapted since 2016 and prohibits the sale of WPs by mail order to minors, is an important step to reduce the illegal sale to minors. A consistent and frequent age control in shisha bars should continue to be carried out by public authorities. Information campaigns (also for parents) may help to decrease the private use of WPs. The ban on marketing tobacco with characteristic flavors (e.g., menthol) implemented in Germany by § 5 of the Tobacco Products Act in 2020 is an important step to prevent young people from consuming flavored WP tobacco. A further ban on the advertising of tobacco products or combined warnings (consisting of pictures and text) are planned for WP tobacco in Germany from May 2024, which will help to increase the awareness about health hazards connected to WP use. Further research should explore why many adolescent WP users do not see themselves as smokers, that is, the beliefs and motives that underlie this. More research is needed on the consumption patterns (e.g., number of puffs, duration of a WP session) and on the type of WP use (e.g., types of WP tobacco, use of charcoal). Moreover, the association between the consumption of WP and other substances, e.g., cigarettes, should be investigated more in detail.

## Figures and Tables

**Figure 1 ijerph-17-07740-f001:**
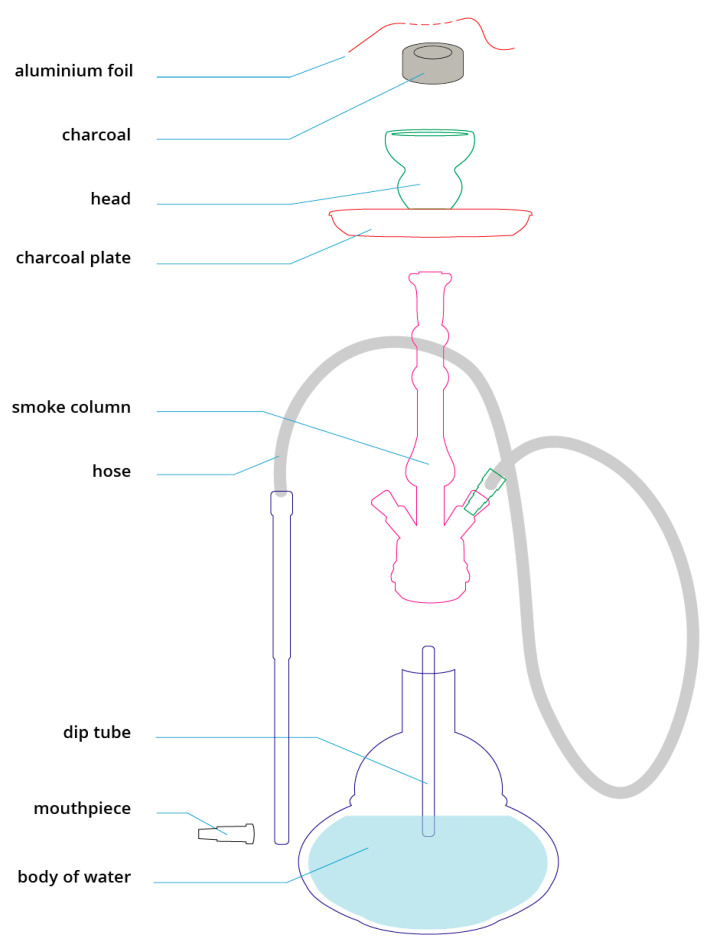
Components of a waterpipe.

**Figure 2 ijerph-17-07740-f002:**
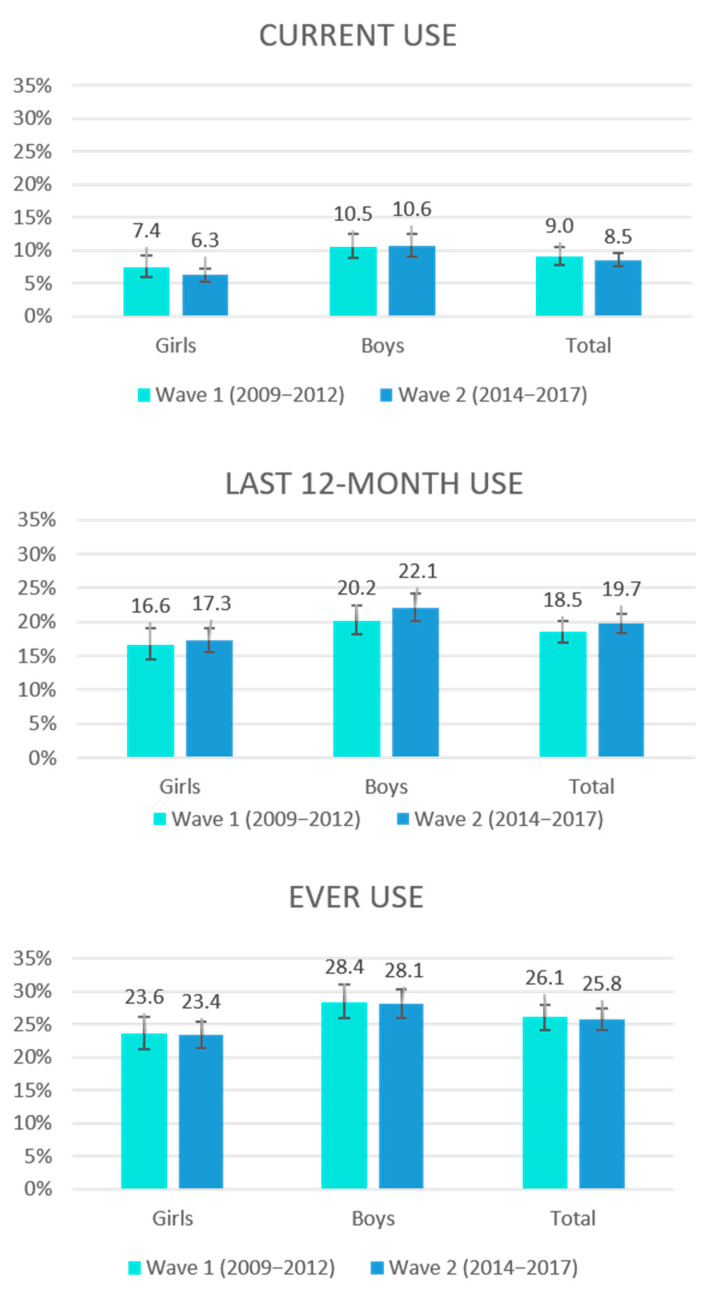
Prevalence of waterpipe use over time among 11- to 17-year-olds in Germany (weighted data).

**Table 1 ijerph-17-07740-t001:** Prevalence of different waterpipe (WP) usage patterns, stratified by sociodemographic characteristics and smoking status (weighted data).

Characteristics	Current WP User ^♦^ *n* = 446		Last 12-Month WP User *n* = 1101		Ever WP User *n* = 1415	
	%	95% CI	%	95% CI	%	95% CI
Total (Girls and Boys)	8.5	(7.5–9.6)	19.7	(18.3–21.2)	25.8	(24.2–27.5)
**Girls**	**6.3**	**(5.3–7.3)**	**17.3**	**(15.6–19.1)**	**23.4**	**(21.4–25.5)**
Age, years						
11	−	−	0.1	(0.0–0.7)	0.7	(0.2–3.0)
12	0.6	(0.1–3.2)	1.5	(0.6–3.5)	3.7	(2.0–6.8)
13	0.7	(0.3–1.5)	6.3	(3.9–9.9)	8.7	(5.7–12.9)
14	6.1	(3.9–9.4)	15.5	(12.1–19.7)	18.9	(14.9–23.6)
15	9.5	(6.6–13.5)	22.1	(17.4–27.6)	28.3	(22.9–34.4)
16	12.1	(8.9–16.2)	36.6	(31.1–42.5)	46.1	(40.1–52.3)
17	12.9	(9.6–17.1)	33.8	(28.2–40.0)	50.6	(44.2–56.9)
Socioeconomic status *						
Low	7.0	(4.5–10.8)	20.5	(16.2–25.5)	30.2	(25.0–36.0)
Middle	6.8	(5.6–8.2)	17.4	(15.4–19.6)	23.1	(20.6–25.7)
High	4.1	(2.6–6.2)	12.9	(10.0–16.5)	16.2	(13.0–20.1)
Education ^≠^						
Secondary/Middle/Comprehensive school	6.6	(5.1–8.4)	18.8	(16.4–21.5)	25.1	(22.1–28.3)
Technical/Academic secondary school	5.9	(4.6–7.5)	15.5	(13.4–18.0)	21.5	(18.9–24.3)
Migration background °						
Yes, one-sided	6.7	(3.8–11.6)	19.0	(14.0–25.3)	23.1	(17.5–29.8)
Yes, both-sided	3.7	(2.0–6.8)	11.2	(7.8–15.8)	18.1	(13.6–23.7)
No	6.8	(5.7–7.9)	18.4	(16.5–20.5)	24.7	(22.5–26.9)
Current smoking status						
Smoker	35.1	(28.0–42.9)	73.8	(64.9–81.1)	86.3	(77.6–92.0)
Non-smoker	4.2	(3.3–5.2)	13.2	(11.5–15.0)	18.9	(17.0–21.1)

**Boys**	**10.6**	**(9.1–12.4)**	**22.1**	**(20.2–24.2)**	**28.1**	**(25.9–30.3)**
Age, years						
11	−		1.7	(0.7–4.2)	2.0	(0.9–4.4)
12	0.5	(0.2–1.6)	2.4	(1.2–4.5)	3.1	(1.7–5.5)
13	3.9	(1.8–8.1)	8.2	(5.3–12.4)	12.0	(8.6–16.5)
14	5.7	(3.4–9.5)	17.6	(13.6–22.5)	22.4	(17.8–27.7)
15	12.4	(8.7–17.2)	29.8	(24.1–36.3)	38.3	(32.0–45.0)
16	21.5	(15.4–29.1)	37.9	(31.0–45.3)	47.9	(41.4–54.5)
17	26.4	(20.9–32.8)	50.3	(43.5–57.2)	61.6	(54.9–67.9)
Socioeconomic status *						
Low	11.6	(7.1–18.3)	22.5	(16.7–29.6)	28.2	(21.7–35.6)
Middle	10.5	(8.7–12.6)	22.9	(20.4–25.6)	29.1	(26.3–32.2)
High	10.7	(8.0–14.2)	19.1	(15.9–22.9)	24.1	(20.6–28.0)
Education ^≠^						
Secondary/Middle/Comprehensive school	9.8	(7.6–12.5)	21.1	(18.3–24.1)	27.2	(23.8–30.9)
Technical/Academic secondary school	10.4	(8.1–13.1)	21.9	(18.6–25.5)	27.5	(24.4–30.8)
Migration background °						
Yes, one-sided	17.9	(11.0–27.8)	30.4	(22.4–39.9)	36.3	(27.9–45.7)
Yes, both-sided	11.5	(7.2–17.8)	19.9	(14.2–27.1)	27.9	(21.5–35.4)
No	9.7	(8.2–11.4)	21.7	(19.6–23.9)	27.2	(24.9–29.6)
Current smoking status						
Smoker	46.8	(37.2–56.7)	84.5	(75.4–90.7)	93.5	(87.5–96.7)
Non-smoker	8.2	(6.7–10.0)	18.1	(16.2–20.2)	23.7	(21.5–26.1)

The numerator for the calculation refers to the total number in the corresponding series (e.g., 50.6% of 17-year-old girls report WP ever use). Bold printed indicates the prevalence for the respective group. **^♦^** Defined as using WP in the last 30 days. ° One-sided indicates children and adolescents having one parent not born in Germany or without German citizenship; two-sided indicates children and adolescents who themselves migrated to Germany and have at least one parent who was not born in Germany, and children and adolescents whose parents were both born in a country other than Germany or non-German nationals. * Socioeconomic status generated as a household characteristic based on parental levels of education, occupational status, and income. ^≠^ German equivalents to school types: Secondary school = Hauptschule; Middle school = Realschule; Comprehensive school = Gesamtschule; Technical secondary school = Fachoberschule (FOS); Academic secondary school = Gymnasium.

**Table 2 ijerph-17-07740-t002:** Multivariable associations between different waterpipe (WP) use patterns and sociodemographic characteristics and smoking status.

Covariates ^†^	OR (95% CI)											
	Model (I) Current WP Use ^♦^ vs. Never WP Use				Model (II) Last 12-Month WP Use vs. Never WP Use				Model (III) Ever WP Use vs. Never WP Use			
	**Girls**		**Boys**		**Girls**		**Boys**		**Girls**		**Boys**	
Age, Years ^>^	1.97 ***	(1.69–2.29)	2.20 ***	(1.92–2.52)	1.94 ***	(1.77–2.14)	1.92 ***	(1.74–2.13)	1.91 ***	(1.75–2.08)	1.91 ***	(1.74–2.10)
Socioeconomic status ^‡^												
Low	1.60	(0.71–3.58)	0.72	(0.31–1.70)	1.27	(0.74–2.17)	0.95	(0.53–1.69)	1.66 *	(1.02–2.71)	0.92	(0.54–1.57)
Middle	1.13	(0.59–2.18)	0.76	(0.45–1.28)	0.99	(0.66–1.48)	1.06	(0.73–1.55)	1.13	(0.79–1.61)	1.10	(0.77–1.58)
High (ref)	1		1		1		1		1		1	
Education ^≠^												
Secondary/Middle/Comprehensive school	1.56	(1.00–2.45)	1.48	(0.91–2.41)	1.82 ***	(1.32–2.51)	1.27	(0.90–1.78)	1.63 **	(1.22–2.17)	1.28	(0.95–1.72)
Technical/Academic secondary school (ref)	1		1		1		1		1		1	
Migration background °												
Yes, one-sided	0.98	(0.43–2.23)	3.03 **	(1.36–6.77)	0.91	(0.50–1.65)	2.04 *	(1.18–3.54)	0.79	(0.46–1.33)	2.01 **	(1.23–3.27)
Yes, both-sided	0.34 *	(0.15–0.80)	1.19	(0.55–2.61)	0.38 ***	(0.22–0.65)	0.87	(0.49–1.55)	0.44 ***	(0.27–0.70)	0.90	(0.56–1.46)
No (ref)	1		1		1		1		1		1	
Current smoking status												
Smoker	48.27 ***	(24.12–96.59)	67.57 ***	(18.02–253.32)	28.03 ***	(14.26–55.08)	40.60 ***	(15.38–107.13)	21.55 ***	(10.99–42.28)	33.45 ***	(13.10–85.40)
Non-smoker (ref)	1		1		1		1		1		1	

^†^ All listed covariates were included in models I–III. Data are presented as odds ratio (OR) 95% confidence interval (CI). * *p* < 0.05; ** *p* < 0.01; *** *p* < 0.001. ^>^ Age was treated as a continuous variable in the regression analyses. ° One-sided indicated children having one parent not born in Germany or without German citizenship; both-sided indicates children who themselves migrated to Germany and have at least one parent who was not born in Germany and children and adolescents whose parents were both born in a country other than Germany or non-German citizens. ^♦^ Defined as using WP in the past 30 days. ^‡^ Socioeconomic status generated as a household characteristic based on parental levels of education, occupational status, and income. ^≠^ German equival ents to school types: Secondary school = Hauptschule; Middle school = Realschule; Comprehensive school = Gesamtschule; Technical secondary school = Fachoberschule (FOS); Academic secondary school = Gymnasium.

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
