# Peer review of "Waterpipe Use among Adolescents in Germany: Prevalence, Associated Consumer Characteristics, and Trends (German Health Interview and Examination Survey for Children and Adolescents, KiGGS)"

_ijerph, 2020, doi:10.3390/ijerph17217740_

Round 1

Reviewer 1 Report

The paper presents a solid analyses on an important theme, using a large national-level data set. The text is well-written but could be improved in a number of respects.

  1. The overall prevalence rates are strongly determined by the age range covered by the study. It might be informative, also from a life course perspective, to give greater emphasis to the prevalence rates as observed at the ‘final’ age of 17 years. The high prevalence rates at this age (33% last-year use for girls, even 50% for boys) get obscured by the inclusion of young ages where WP is naturally not common. I recommend specific attention to 16-17 year olds in the paper, including the Abstract.

  1. The final sentence of the Abstract, about “specific prevention strategies” is not backup by a discussion in the main text of what kind of strategies these could be. The discussion is most clear about policies that are population wide (universal), instead being targeted or tailored (particularistic). Recommendations for policies thus need more reflection.

  1. The summary of results in both the Abstract and the Discussion is too simple with regards to educational level and migration background. The authors should distinguish between the sometimes opposing patterns for girls and boys, or say that there is not one straightforward pattern.

  1. The discussion section lacks an attempt to help the reader to understand the results that are observed. E.g. why do girls and boys differ in the relationship of WP with SES and migration background – does this reflect a similar gender difference in cigarette use until about the 1980s? The current version only repeats the results, and compares these with previous studies. I recommend a reflection of a few results that are key to the paper and/or least expected.

  1. The results regarding “considered themselves smokers” does not neatly fit in the rest of the results, they do not address the research questions of the paper, and they do not lead to novel insights. Moreover, they raise new questions, e.g. how does “considered to be smoker” relate to actual smoking status? Given this all, I would remove these results.

A minor comment:

  1. The rationale of classification of education into two levels is not obvious for someone who is not familiar with the German educational system. It this hierarchical, and why? This needs more clarification.

Author Response

Response to Reviewer 1 Comments

The paper presents a solid analyses on an important theme, using a large national-level data set. The text is well-written but could be improved in a number of respects.

Point 1: The overall prevalence rates are strongly determined by the age range covered by the study. It might be informative, also from a life course perspective, to give greater emphasis to the prevalence rates as observed at the ‘final’ age of 17 years. The high prevalence rates at this age (33% last-year use for girls, even 50% for boys) get obscured by the inclusion of young ages where WP is naturally not common. I recommend specific attention to 16-17 year olds in the paper, including the Abstract.

Response 1: Thank you very much for the comment. As the use of WP can already be observed in adolescents under the age of 16, we defined the age group to be investigated for our study as 11-17 years in order not to lose this information. However, we also reported the prevalence rates of each individual age group in Table 1. To make the figures in this table more present, we would like to take your advice to pay special attention to the ‘main user group’ of 16-17-year-olds.

Revision: We have added this information in different sections of abstract, results and discussion.

Page 1, lines 22-23: ‘High prevalence rates were particularly found among 16-17-year-old girls and boys.’

Page 5, lines 207-209: ‘12.9% (95% CI = 9.6-17.1) of the 17-year-old girls reported current WP use and 33.8% (95% CI = 28.2-40.0) in the last 12 months. For boys these rates were 26.4% (95% CI = 20.9-32.8) and 50.3% (95% CI = 43.5-57.2), respectively.’

Page 12, lines 255-256: ‘The use of WP seems to be most common in the age group of 16-17-year-olds.’

Point 2: The final sentence of the abstract, about “specific prevention strategies” is not backup by a discussion in the main text of what kind of strategies these could be. The discussion is most clear about policies that are population wide (universal), instead being targeted or tailored (particularistic). Recommendations for policies thus need more reflection.

Response 2: Thank you very much for the hint. The topic of prevention strategies and possible political measures opens a broad field and can only be seen in the context of this article with a short overview.

Revision: We hope to address this point now more satisfactorily by the following concretisation.

Page 13, lines 332-335: ‘In addition to preventive programmes within schools, it would be also useful to provide information about the health hazards and addiction of WP use in sports clubs or on preferred social networks (e.g., YouTube, Facebook, Twitter) visited by young people. A targeted gender-specific approach could also be made here.’

Point 3: The summary of results in both the Abstract and the Discussion is too simple with regards to educational level and migration background. The authors should distinguish between the sometimes opposing patterns for girls and boys, or say that there is not one straightforward pattern.

Response 3: Thank you for this hint.

Revision: We have made the following addition to the discussion section.

Page 12, lines 258-259: ‘We found positive associations of WP use with older age, male sex, migration background, lower educational level, and current smoking status.’

Page 12, lines 259-263: ‘Concerning the associations between WP consumption and education level or migration background, sometimes an opposing relationship was observed for both genders. As shown in Table 2, the association between lower educational level and the use of WP was more pronounced among girls, whereas the association between the migration background and t use of the waterpipe is found primarily among boys.’

Point 4: The discussion section lacks an attempt to help the reader to understand the results that are observed. E.g. why do girls and boys differ in the relationship of WP with SES and migration background – does this reflect a similar gender difference in cigarette use until about the 1980s? The current version only repeats the results and compares these with previous studies. I recommend a reflection of a few results that are key to the paper and/or least expected.

Response 4: You are right; the result that girls and boys are differ in the relationship of WP with SES and migration background has not been adequately discussed. According to intense literature research, we do not know any studies that research the cultural aspects of the consumption of WP by young people in a gender-differentiated way. If you are familiar with the literature on this issue, we would be pleased if you could share it with us, as this is an interesting issue that seems to be under-researched so far. A study from Germany in 2005 [1] found similar conclusions for adults (over 18 years): Among male migrants, the proportion of smokers was higher compared to smokers without a migration background (44.1% vs. 36.5%). The opposite trend was found among women (27.7% vs. 31.7%). Since we can only assume at this point to what extent family structures or the spread of women's tobacco consumption in their countries of origin play a role in this context, and since we cannot provide sufficient scientific evidence to support this, we not attempted to explain this.

Revision: We have made additions to the discussion section.

Page 12, line 296: ‘Similar results have also been described for smoking adults (over 18 years) in Germany [49].’

Point 5: The results regarding “considered themselves smokers” does not neatly fit in the rest of the results, they do not address the research questions of the paper, and they do not lead to novel insights. Moreover, they raise new questions, e.g. how does “considered to be smoker” relate to actual smoking status? Given this all, I would remove these results.

Response 5: Thank you very much for bringing this to our attention. We would like to explain the background of this issue from our point of view. We think it is of central importance whether a young person who uses a WP perceives himself as a smoker or non-smoker. This can also lead to an underestimation of the health risks of WP consumption, as mentioned at various points in the article. As there is no current data available from Germany, we can contribute with our figures to learn more about this and use this knowledge to adapt prevention strategies.

Revision: We have added this in the introduction section.

Page 3, lines 105-108: ‘There are also no current data for Germany, about the percentage of WP users who perceive themselves as smokers or non-smokers. This is an important issue, which can influence the perception of health risks of WP tobacco consumption and the creation of prevention programmes.’

Page 4, line 112: ‘(…) and the self-assessed smoking status (…)’

Point 6: The rationale of classification of education into two levels is not obvious for someone who is not familiar with the German educational system. It this hierarchical, and why? This needs more clarification.

Response 6: Thank you for your request.

Due to its federal structure, there is no uniform school system in Germany. Because some federal states now have a two-tier school system, we decided to report only on the categories "Real-, Haupt-, Gesamtschule" vs. "FOS/Gymnasium". Persons who have already finished school were assigned according to their qualifications.

Revision: We have added this information in the measurements section.

Page 5, lines 161-163: ‘Due to its federal structure, there is no uniform school system in Germany. Because some federal states now have a two-tier school system, we have decided (…) for the following analyses, to define two groups (…)’

References:

[1] Lampert, T.; Burger, M. Distribution and patterns of tobacco consumption in Germany [Verbreitung und Strukturen des Tabakkonsums in Deutschland]. Bundesgesundheitsblatt – Gesundheitsforschung – Gesundheitsschutz 2005, 48, 1231–1241, doi.org/10.1007/s00103-005-1158-7.

Reviewer 2 Report

The aim of this study was to examine prevalence of waterpipe consumption among German adolescents. In my opinion, the article presents an interesting topic, uses appropriate statistical analysis with a big sample and presents interesting results.

However, I think that an important issue regarding the objective should be edited before publishing. The aim of the study is not explicitly explained in the abstract. Moreover, in the introduction authors state that the aim is to "investigate the prevalence of waterpipe consumption among 11- to 17-year-old boys and girls, to describe the frequency of waterpipe use, examine the associations between SES of the family of origin, migration background, and waterpipe consumption among adolescents, and to monitor changes in prevalence between the first follow-up of the KiGGS study (KiGGs Wave 1, 2009-2012) and KiGGS Wave 2 (2014-2017). These data can help in the identification of different risk profiles to develop targeted group-specific and gender-sensitive prevention strategies". I  find the aim too detailed to be in the introduction. I think in the introduction the aim should be stated in a simple way and then offer all this detailed information in the methods section.

Author Response

Response to Reviewer 2 Comments

The aim of this study was to examine prevalence of waterpipe consumption among German adolescents. In my opinion, the article presents an interesting topic, uses appropriate statistical analysis with a big sample and presents interesting results.

Point 1: However, I think that an important issue regarding the objective should be edited before publishing. The aim of the study is not explicitly explained in the abstract. Moreover, in the introduction authors state that the aim is to "investigate the prevalence of waterpipe consumption among 11- to 17-year-old boys and girls, to describe the frequency of waterpipe use, examine the associations between SES of the family of origin, migration background, and waterpipe consumption among adolescents, and to monitor changes in prevalence between the first follow-up of the KiGGS study (KiGGs Wave 1, 2009-2012) and KiGGS Wave 2 (2014-2017). These data can help in the identification of different risk profiles to develop targeted group-specific and gender-sensitive prevention strategies". I find the aim too detailed to be in the introduction. I think in the introduction the aim should be stated in a simple way and then offer all this detailed information in the methods section.

Response 1: Thank you very much for your positive feedback and advice. We think that the aims in the introduction section are correctly placed in this length. If the editor will have placed it somewhere else, we will, of course, be pleased to accept it.

Revision: We have added the aims of the study in the abstract and have adapted the sentence regarding the aims in the introduction section to make it easier to read

Page 1, line 15: ‘We aimed to describe (…)’

Page 3,4 lines: 108-114: ‘We therefore aimed to WP use and associated factors among German adolescents. More specifically (…) and, the self-assessed smoking status, examine the associations between sociodemographic factors, smoking status and WP consumption among adolescents, and to monitor trends between the previous and the current wave of the KiGGS study.’

Reviewer 3 Report

REVIEW OF Manuscript ID: ijerph-949064 There are several potential problems with subject manuscript that need clarification. First, the target of the manuscript is waterpipe use by youth aged 11 to 17 in Germany. Is purchase of waterpipe tobacco, hookahs, and accessories to such persons permitted under German law? If not, should not the discussion be focused on steps to prevent illegal sales and other distributions to those below the legal age for use of waterpipe tobacco, hookahs, and accessories? Also, what about entry to hookah bars and other points of access to waterpipe smoking? Second, the study-period for the data reported in the manuscript is 2014 through 2017. During this time period the German rules on the composition of shisha tobacco changed. In particular, the limit on glycerin content of 5% of product weight was removed. Analytical data for some contemporary products show ~50% glycerin, ~10% water, ≥3% propylene glycol. This the shisha does not burn. It is only heated. Thus, the authors are likely comparing usage data on products that changed during the study period. Not only did the products change, the composition of the emissions changed with the harmful compounds in the emissions reduced only to those coming from the burning charcoal used to heat the shisha tobacco. Third, if the authors were studying use of nonmenthol cigarettes manufactured by the major international cigarette companies selling in German (e.g., PMI, BAT, JTI, Imperial Brands), the manuscript would be very acceptable after discussing the legal age of purchase in Germany and the sources of product for underage smokers. Manufactured cigarettes tend to be a very homogenous group of products, and the patterns of smoking behavior are well known. This is definitely not the case for shisha (waterpipe) tobaccos, the waterpipes used to heat the tobaccos, the accessories used, and how all those components are assembled into a smoking system. Moreover, how available are correctly assembled systems available to youth? Furthermore, is a “use” a puff or two, or is it an entire smoking session? In addition, are there are other confounding variables that would enhance or detract from adolescents propensity to begin or continue waterpipe smoking? Hopefully, the authors will address these concerns in their revised document.

Author Response

Response to Reviewer 3 Comments

There are several potential problems with subject manuscript that need clarification.

Point 1: First, the target of the manuscript is waterpipe use by youth aged 11 to 17 in Germany. Is purchase of waterpipe tobacco, hookahs, and accessories to such persons permitted under German law? If not, should not the discussion be focused on steps to prevent illegal sales and other distributions to those below the legal age for use of waterpipe tobacco, hookahs, and accessories? Also, what about entry to hookah bars and other points of access to waterpipe smoking?

Response 1: Thank you for taking up this important point. In Germany the purchase and consumption of waterpipes, waterpipe tobacco and accessories is prohibited by law for persons under 18. This also includes the visit of shisha bars.

Revision: We have added the following information regarding the legal situation in Germany to the introduction and the conclusion section.

Page 3, lines 74-85: ‘In Germany, there are legislative measures at both state and federal level to regulate WP consumption (Bundesnichtraucherschuztgesetz (‘Federal Non-Smoker Protection Act’), Jugendschutzgesetz (‘Youth Protection Act‘), Tabakerzeugnisgesetz (‘Tobacco Products Act’), Nichtraucherschutzgesetz (‘Non-smoker protection law’)). The German Tobacco Products Act regulates ingredients, emission levels and information requirements for tobacco and related products. In 2016 the ingredients of WP tobacco changed. The youth protection act regulates the distribution of tobacco products. In 2007, the age limit for the consumption of tobacco products has been raised to 18 years (previously this was allowed at 16). It is not permitted to sell tobacco products to minors. Children and adolescents under the age of 18 are not allowed to smoke in publicly accessible rooms in places open to the public and otherwise in public places. These measures were accompanied by a tobacco prevention programme. Purchase of waterpipe tobacco and accessories or the entry to a shisha bar are not permitted to people under 18 years of age.’

Page 13, lines 335-342: ’The law for the protection of youth, which has been adapted since 2016 and prohibits the sale of WPs by mail order to minors, is an important step to reduce the illegal sale to minors. A consistent and frequent age control in shisha bars should continue to be carried out by public authorities. The private use of WPs should be reduced with information campaigns (also for parents). The ban on marketing tobacco with characteristic flavours (e.g. menthol) implemented in Germany by § 5 of the Tobacco Products Act - since 20 May 2020 - is a further important step towards preventing young people from consuming the flavoured shisha tobacco.’

Point 2: Second, the study-period for the data reported in the manuscript is 2014 through 2017. During this time period the German rules on the composition of shisha tobacco changed. In particular, the limit on glycerin content of 5% of product weight was removed. Analytical data for some contemporary products show ~50% glycerin, ~10% water, ≥3% propylene glycol. This the shisha does not burn. It is only heated. Thus, the authors are likely comparing usage data on products that changed during the study period. Not only did the products change, the composition of the emissions changed with the harmful compounds in the emissions reduced only to those coming from the burning charcoal used to heat the shisha tobacco.

Response 2: Many thanks for the comment. This is an important point, especially concerning health risks. Thus, the changes to the product you mentioned could not be included in the survey during the survey phase.

Revision: We now note the change in the ingredients (see Revision to point 1) and in the discussion part.

Page 13, lines 311-314: ‘To be able to assess the health risks arising from the consumption of WP tobacco, the ingredients of WP tobacco play an integral part. Unfortunately, the composition of WP tobacco or the number of puffs during a session could not be investigated in this study.’

Point 3: Third, if the authors were studying use of nonmenthol cigarettes manufactured by the major international cigarette companies selling in German (e.g., PMI, BAT, JTI, Imperial Brands), the manuscript would be very acceptable after discussing the legal age of purchase in Germany and the sources of product for underage smokers. Manufactured cigarettes tend to be a very homogenous group of products, and the patterns of smoking behavior are well known. This is definitely not the case for shisha (waterpipe) tobaccos, the waterpipes used to heat the tobaccos, the accessories used, and how all those components are assembled into a smoking system. Moreover, how available are correctly assembled systems available to youth? Furthermore, is a “use” a puff or two, or is it an entire smoking session? In addition, are there are other confounding variables that would enhance or detract from adolescents propensity to begin or continue waterpipe smoking? Hopefully, the authors will address these concerns in their revised document.

Response 3: We have already addressed some of the issues in your first comment. We also see the point of defining use as very important. Of course, it makes a difference whether the respondents take a puff or take part in a longer lasting waterpipe session. Unfortunately, we were not able to go into this differentiation in our survey.

Revision: We have added this point as a limitation of the study in the discussion section (see revision to point 2) and added this as new aspects for further research in the conclusion section.

Page 13, lines: 346-348: ‘Furthermore, more research is needed on the consumption pattern (e.g., number of puffs, duration of a WP session) and the kind of WP use (e.g., types of WP tobacco, use of charcoal).’

Reviewer 4 Report

The paper describes a study that reports the prevalence and correlates of waterpipe use among German adolescents. Such descriptive epidemiological papers seldom yield exciting, new insights, but when well-conducted they do yield important information for monitoring secular trends and identifying risk or protective factors that might inform policy and intervention. The topic of waterpipe use among youth is one of public health import given relatively high rates of use documented across several countries. This study has the desirable characteristic for descriptive epidemiological research that the data are based on a sufficiently large probability sample. There are two primary areas that need addressing. The first is to more clearly highlight the unique contribution that this study makes. The authors note other population-based studies that have examined waterpipe use among German adolescents. This is not intended to mean that there is no value in having multiple similar studies. Replication research is underappreciated and much needed. However, this study could be clearer on what its unique contributions to the literature are and what is intended to be a replication of prior research. Second, greater clarity on the sampling methodology and analytic approach is needed. While I appreciate the desire to not repeat details that have been published elsewhere, I had difficulty locating at least one of the references. Further, issues related to the recruitment of migrants and changing modes of survey administration were first mentioned in the limitations section of the discussion section. Related and other, generally minor, issues are outlined below:

  1. Methods
    1. It is not clear why respondents were selected at random for the current study. Why not use the entire sample of KiGGS W2 participants?
    2. It should be clearly indicated that this is a cross-sectional study. With language such as “baseline” “first follow-up” and “second follow-up”, readers could be confused that this is a longitudinal study. That is unless I am mistaken, and this is a longitudinal study. Either way, this should be explicitly stated.
    3. As noted above, please better describe the sampling and data collection methodology. Note the modes of data collection for waves 1 and 2 data used in this study. Please provide AAPOR response metrics for both waves and any pertinent details on the weighting and representativeness of the samples, such as with the migrant status of the participants mentioned in the discussion section.
    4. Were pics of waterpipes provided in the survey? Its possible some youth have used waterpipe in social settings but not known what it was called. In my experience, pictures improve tobacco product identification, address differing terminology users may have for a tobacco product, and improve measurement.
    5. Was e-cigarette use measured? If so, please incorporate as a predictor of waterpipe use. I’m not certain about Germany, but in some countries (e.g., U.S.A.), e-cigarette use is the most commonly used tobacco product among youth.
    6. Please clarify the amount of missing data, whether analyses were conducted to identify systematic patterns of missingness and the results of such analyses, and whether missingness (leading to removal from the analysis) was addressed in the weighting.
  2. Results
    1. Line 167 “Respondents with migration background and current smokers…” – whereas percentages are reported elsewhere, this sentence is missing estimated percentages.
    2. Table issues: (a) in Table 1, please format to be clearer that the sociodemographic and smoking breakdowns are stratified by gender. It seems that rows pertaining to “Boys” are colored blue, but this was not so for “Girls”. This would be fine, but I suggest avoiding the blue/pink stereotype. Rather, one could use varying degrees of row header indentation and/or boldface to denote the sub-breakdown within gender. (b) Table 1 – align so that CI is closer to their associated % than with percent for other outcomes. (c) Table 1 – please explain how the range for current, past 12 months, and ever use percentages by school types for boys do not contain the overall %. For instance, the current use estimate for boys is 10.6% but for boys in secondary/middle/comprehensive school it’s 5.9% and for 6.6%. This doesn’t make sense if these school types are mutually exhaustive exclusive.
    3. Figure 1 – While the figure provides a nice visual, a table would provide more precise information about the estimates (exact estimates and CI limits).
  3. Discussion
    1. Lines 242-243 “higher ORs for current and ever use among respondents who were male, older, and who had a migration background” – this could be misinterpreted in that it only applies to males who have a one-sided migration background. The explanation for these results (“greater amount of missing data for this group”), while the female group exhibited a different pattern, needs to be better explained. How much missing data were there for this group, what was the nature of the missing data, and why did it occur. And how could it explain the higher rate of use for males one-sided migration but not both-sided migration, where a different pattern was observed for females? Could the different pattern at wave 1 be due to a different pattern of migration then (with a different country of origin distribution)? These results are counterintuitive and need a more nuanced discussion of the possible explanations, including and varying patterns of migration between waves and genders or selection bias or nonrandom missingness.
    2. Particularly for readers outside of Germany, could the authors provide relevant policy context pertaining to the regulation and enforcement of restrictions prohibiting the sale of and youth possession of tobacco and, in particular, waterpipe.

Author Response

.

Round 2

Reviewer 1 Report

The authors have addressed all my comments to the previous paper. In some cases, I thought that the authors could have incorporated my comments to a greater extent. However, I accept that their choices, as the new version of the paper is sound and sufficiently attractive.

Author Response

Many thanks for your positive feedback. We are pleased that you appreciate the current version.

Reviewer 4 Report

The authors were responsive to the initial reviews and the manuscript is improved. There are only a few issues (mostly minor) remaining:

  1. I appreciate the additional detail on the methods and appreciate the disinclination to provide greater detail that has already been published. However, the publications cited to find additional methodological detail is either mostly behind a paywall or not in the same language as this publication. Given this and the recognized importance of reporting AAPOR sample response metrics (and the issues raised about sample representativeness with respect to immigrant populations), I'd strongly encourage providing greater detail, especially response metrics.
  2. Regarding missingness - I appreciate the detail. I suggest that the authors add an abbreviated version of their response. For instance, what was the amount of missing data, the key results from analyses to examine whether missingness was systematic, and steps taken (or not taken) to address missingness in the analyses.
  3. The revised text has several typographical (often missing spaces between words and sentences) and some grammatical errors. For instance, line 108 "We therefore aimed to WP use" seems to be missing words." I suggest a careful proofread by a co-author with high proficiency in written English.
  4. I appreciate the regulatory and policy context pertaining to youth waterpipe use in Germany. This paragraph was informative, but it did read rather disjointed. For instance, "In 2016 the ingredients of WP tobacco changed." -- this seems out of place and needs elaboration on why and how ingredients changed.

Author Response

Point 1: I appreciate the additional detail on the methods and appreciate the disinclination to provide greater detail that has already been published. However, the publications cited to find additional methodological detail is either mostly behind a paywall or not in the same language as this publication. Given this and the recognized importance of reporting AAPOR sample response metrics (and the issues raised about sample representativeness with respect to immigrant populations), I'd strongly encourage providing greater detail, especially response metrics.

Response 1: Thank you very much for your comment. The publications we cite are all free available and can be find mostly in English.

Revision: We have now added the DOI to make it easier to find the references. Here the requested details can be read in English:

Page 17, line 495: ‘[38] Hoffmann, R.; Lange, M.; Butschalowsky, H.; Houben, R.; Schmich, P.; Allen, J.; Kuhnert, R.; Schaffrath Rosario, A.; Gößwald, A. KiGGS Wave 2 cross-sectional study – participant acquisition, response rates and representativeness. J Health Monit 2018, 3, 78–91, doi: 10.17886/RKI-GBE-2018-032.’

Page 17, line 498: ‘[39] Frank, L.; Yesil-Jürgens, R.; Born, S.; Hoffmann, R.; Santos-Hövener, C.; Lampert, T. Improving the inclusion and participation of children and adolescents with a migration background in KiGGS Wave 2. J Health Monit 2018, 3, 126–142, doi: 10.17886/RKI-GBE-2018-034.’

We are pleased to follow your remark and have added the following sentences in the methods section:

Page 4, lines 133-137: ‘The response rate (according to AAPOR response rate 2) of KiGGS Wave 2 was 40.1% in total [38]. A multi-step approach was used to include people with a migration background in KiGGS Wave 2. The share of children and adolescents of non-German nationality in KiGGS Wave 2 corresponds to the population figures from the Federal Statistical Office [39].’

Page 4, line 149: ‘(…) (response rate 40.1%).’

Point 2: Regarding missingness - I appreciate the detail. I suggest that the authors add an abbreviated version of their response. For instance, what was the amount of missing data, the key results from analyses to examine whether missingness was systematic, and steps taken (or not taken) to address missingness in the analyses.

Response 2: Thank you for the advice.

Revision: We have added the following information’s regarding missingness to the results and discussion section.

Page 5, lines 206-210: ‘The pattern of missing values showed a higher amount of missing values among boys with migration background, boys with lower SES and lower education level, and among girls with lower SES and multivariable analyses showed that the odds of missing values are especially high among boys with a both-sided migration background (data not shown).’

Page 13, lines 301f.: ‘Migration background is a known correlate of WP use described in previous KiGGS Waves and other studies.’

Page 13, lines 307-309: ‘Hence, we speculate that the particular high amount of missing values among boys with a both-sided migration background might explain their low prevalence of WP use. One explanation for these results may be the greater amount of missing data for this group in KiGGS Wave2.’

Page 14, lines 336f.: ‘Systematic identification of patterns of missing items was not feasible, but could help to interpret results more accurately in further studies.’

Point 3: The revised text has several typographical (often missing spaces between words and sentences) and some grammatical errors. For instance, line 108 "We therefore aimed to WP use" seems to be missing words." I suggest a careful proofread by a co-author with high proficiency in written English.

Response 3: Thank you for your careful reading. We now hope to have improved the typographical and grammatical errors.

Revision: We have corrected the text for grammatical and typographical errors in several parts. We have made no changes in content.

Point 4: I appreciate the regulatory and policy context pertaining to youth waterpipe use in Germany. This paragraph was informative, but it did read rather disjointed. For instance, "In 2016 the ingredients of WP tobacco changed." -- this seems out of place and needs elaboration on why and how ingredients changed.

Response 4: Thank you very much for the hint. The topic of regulatory and policy context opens a broad field and can only be seen in the context of this article with a short overview. We have included the sentence regarding the change of ingredients on the advice of another reviewer. However, as this passage is not of primary importance to the article, we have only mentioned this hint to allow the reader to do further research. Therefore, this issue is not further explained here. Rules regarding the composition of WP tobacco changed on 20 May 2016 in Germany. In particular, the limit on glycerin content of 5% (of product weight) was removed. The higher degree of moisture in WP tobacco now can influence the smoking experience and the popularity of WP consumption among young people. We would also agree not to add the remark and to delete the sentence ‘In 2016 the ingredients of WP tobacco changed’ if it is perceived here as irritating and misplaced. We are willing to leave this decision to the editor.

Revision: We have added this information.

Page 3, line 79 (…) (% content of glycerin).’
